

# Comprehensive identification and expression analysis of *CAMTA* gene family in *Phyllostachys edulis* under abiotic stress

Ce Liu[1,2] and Dingqin Tang[1,2]

[1] School of Forestry and Biotechnology, Zhejiang A&F University, Hangzhou City, Lin'an, China
[2] State Key Laboratory of Subtropical Forest Cultivation, Zhejiang A&F University, Hangzhou City, Lin'an, China

## ABSTRACT

**Background:** Calmodulin-binding transcription factor (*CAMTA*) is a major transcription factor regulated by calmodulin (CaM) that plays an essential role in plant growth, development and response to biotic and abiotic stresses. The *CAMTA* gene family has been identified in *Arabidopsis thaliana*, rice (*Oryza sativa*) and other model plants, and its gene function in moso bamboo (*Phyllostachys edulis*) has not been identified.

**Results:** In this study, a total of 11 *CAMTA* genes were identified in *P. edulis* genome. Conserved domain and multiplex sequence alignment analysis showed that the structure between these genes was highly similar, with all members having CG-1 domains and some members having TIG and IQ domains. Phylogenetic relationship analysis showed that the *CAMTA* genes were divided into five subfamilies, and gene fragment replication promoted the evolution of this gene family. Promoter analysis revealed a large number of drought stress-related cis-acting elements in *PeCAMTAs*, and similarly high expression of the *CAMTA* gene family was found in drought stress response experiments, indicating the involvement of this gene family in drought stress. Gene expression pattern according to transcriptome data revealed participation of the *PeCAMTA* genes in tissue development.

**Conclusions:** Our results present new findings for the *P. edulis CAMTA* gene family and provide partial experimental evidence for further validation of the function of *PeCAMTAs*.

Corresponding author
Dingqin Tang, tang@zafu.edu.cn

## INTRODUCTION

Calcium ($Ca^{2+}$) ions are involved in many cellular signaling pathways as prevalent secondary messengers in eukaryotes (*Wu et al., 2016*). $Ca^{2+}$-mediated signaling plays a key role in the transmission of signals generated by different stimuli. Thus, it mediates various stress responses in plants (*Evans, McAinsh & Hetherington, 2001*; *White & Broadley, 2003*). CaM can bind to $Ca^{2+}$ as flexible $Ca^{2+}$/CaM structural proteins, and $Ca^{2+}$ in structural proteins can interact with many proteins, allowing CaM in structural proteins to regulate protein targets in many different signaling pathways (*Bouché et al., 2005*; *DeFalco et al., 2016*; *Poovaiah et al., 2013*; *Yamniuk & Vogel, 2004*). $Ca^{2+}$ and CaM complexes

deliver various endogenous and exogenous signals through multiple interactions with transcription factors (TFs) depending on how the plant responds (*Kim et al., 2009*). Calmodulin-binding transcription factor (*CAMTA*), a major transcription factor regulated by calmodulin (CaM), was first identified in tobacco in 2000 (*Yang & Poovaiah, 2000*). The *CAMTA* protein structural domain contains the following functional domains: (1) N-terminal containing a CG-1 DNA binding domain; (2) A TIG structural domain engaged in non-specific DNA binding; (3) Ankyrin repeat sequences responsible for mediating interactions between different proteins; (4) a $Ca^{2+}$-dependent CaM binding domain between the N-terminal and C-terminal; (5) IQ motifs interacting with CaM (*Bähler & Rhoads, 2002*; *Bouché et al., 2002*; *Du et al., 2009*; *Finkler, Ashery-Padan & Fromm, 2007*; *Yang & Poovaiah, 2002*).

*CAMTA* transcription factors have been found to have very important and effective functions in plant growth and development, biotic and abiotic stress responses, and the level of expression of *CAMTAs* in response to different stresses varies between species (*Chung et al., 2020*; *Noman et al., 2021*; *Shkolnik et al., 2019*; *Yue et al., 2015*). The important role of *CAMTA3* gene for *Brassica napus* (cabbage, kale and kale type oilseed rape) in cold and disease resistance was found (*Luo et al., 2021*). Two genes, *ZmCAMTA4* and *ZmCAMTA6*, were highly expressed in maize (*Zea mays*) under abiotic stress treatment, and cis-element analysis revealed the involvement of *CAMTA* genes in the association between environmental stress and stress-related hormones (*Liu, Hewei & Min, 2021*). *Wei, Xu & Li (2017)* suggested that *Populus trichocarpa CAMTA* genes play an essential role in resistance to cold stress, and showed that woody plants and crops have different *CAMTA* gene expression patterns under abiotic stresses and phytohormone treatments. The land cotton (*Gossypium hirsutum*) *GhCAMTA11* gene is specifically expressed in roots and under heat stress, and *GhCAMTA7* and *GhCAMTA14* are also expressed under drought stress, indicating that the land cotton *CAMTA* gene family is involved in the growth and development process and stress reaction of land cotton (*Zhang et al., 2022*). It was found that the biochemical response of (*Hevea brasiliensis*) *HbCAMTA3* in response to low temperature stress in rubber trees is similar to that of *AtCAMTA3* in *Arabidopsis*, and that the *AtCAMTA3* gene is also involved in salt stress reaction implying that *HbCAMTA3* in rubber trees is functionally diverse (*Lin et al., 2021*). Interestingly, *TaCAMTA* was found to regulate mainly the drought stress response during the seedling stage of wheat, and *TaCAMTA1b-B. 1* plays an essential role in the response to drought stress caused by water deficit in the nursery stage (*Wang et al., 2022*).

*Phyllostachys edulis* belongs to the genus *Phyllostachys* in the family Gramineae, which is widely distributed in China and is an important bamboo resource with the characteristics of strong adaptability, rapid growth, easy reproduction and good timber (*Lin et al., 2002*; *Xu et al., 2022*; *Yang & Li, 2017*).

We comprehensively analyzed the phylogenetic relationships of the *CAMTA* gene family in *P. edulis* and model plants to elucidate their evolutionary relationships. Using available RNA-seq data and qRT-PCR results, we analyzed the expression profile of *PeCAMTA* gene family genes during plant growth and development, as well as the expression of this gene family in response to abiotic stress. In this study, we identified the

*CAMTA* gene family in *P. edulis* in order to provide relevant data support in future plant breeding studies and to open new avenues for further elucidation of its role in *P. edulis* signal transduction.

## MATERIALS AND METHODS

### Identification of *CAMTA* genes in *P. edulis* genome

All files associated with the whole genome sequence data of *Phyllostachys edulis* were downloaded from the database website (http://gigadb.org/dataset/100498). A numerical tabular Hidden Markov Model (Profile HMM) was constructed using HMMER3 (http://hmmer.janelia.org/) to match the *Phyllostachys edulis* protein database (significant E value set to no more than $1 \times 10^{-20}$) (*Finn, Clements & Eddy, 2011*). The *CAMTA* domain (PF03859) obtained from the Pfam database was screened and integrated (*Finn et al., 2016*), and the candidate gene family members were obtained from the initial screening. The *CAMTA* structural domains of the candidate family members were analyzed using SMART (*Letunic, Doerks & Bork, 2012*), along with the Plant TFDB and NCBI BLAST for further comprehensive analysis and identification to obtain candidate *CAMTA* transcription factor families (*Jin et al., 2017*).

### Physicochemical properties and signal peptide analysis of *P. edulis* *CAMTA*

The Sequence Toolkits module of TBtools software (v1.098765) was used to derive the coding sequence (CDS), protein fasta sequence, gene structure and gene location information of *CAMTA* gene family members from the corresponding genome-wide database (*Chen et al., 2020*) using The online tools Prot Param and TargetP 2.0 Server (https://services.healthtech.dtu.dk/service.php?TargetP-2.0) were used to analyze their physicochemical Properties and signal peptides were analyzed.

### Phylogenetic analysis

The whole genome information of *O. sativa*, *Arabidopsis*, *Z. mays*, and *Brachypodium distachyon* was downloaded from the *O. sativa* genome database (https://shigen.nig.ac.jp/rice/oryzabase/), the *Arabidopsis* database (http://www.arabidopsis.org), *Z. mays* database (https://maizegdb.org/), and *B. distachyon* database (https://phytozome-next.jgi.doe.gov/info/Bdistachyon_v3_1), respectively, and based on the obtained *CAMTA* Protein sequences of the four plants, the software ClustalX2.1 was used to contrast the *CAMTA* Protein sequences of *P. edulis*. The sequence alignment results were used to construct phylogenetic trees by the software MEGA7 using the neighbor-joining (NJ) method, and the bootstrap evaluation (Bootstrap) was repeated 1,000 times.

### Analysis of gene structure, motifs, domains

Based on the gene location information of *P. edulis* genome annotation file (GFF), the gene intron and exon sequences were analyzed and the gene structure of *PeCAMTA* gene family was visualized; the NCBI online software CDD was used to forecast the conserved

structural domains of *CAMTA* gene family members, and their amino acid conserved sequences were predicted using the online software MEME (*Bailey et al., 2009*).

## Cis-acting elements in the *PeCAMTA* gene promoter regions

PlantCARE (http://bioinformatics.psb.ugent.be/webtools/plantcare/html/) was used to identify cis-acting elements in the 1,500 bp promoter region upstream of the transcription start site for each gene, and the results were submitted to TBtools (v1.098765) for visualization.

## Chromosome distribution and interspecies covariance analysis

The BLAST module of TBtools (v1.098765) software was used to execution sequence comparison of all proteins in the genome of bamboo, and two-way alignment of *P. edulis* with *O. sativa* and *P. edulis* with *Arabidopsis*, based on genome-wide GFF files, using MC ScanX, Circos (0.69–9) and Multipe Synteny Plot. *CAMTA* gene family chromosome distribution and interspecies covariance were visualized using MC ScanX, Circos (0.69–9) and Multipe Synteny Plot.

## Tissue-specific expression levels of *PeCAMTA* genes

In order to analyze the specific expression of *CAMTA* gene in *P. edulis*, we downloaded RNA-seq data from the NCBI gene expression profiles database (Accession: ERR105067–ERR105076). Transcriptome data which was quantified as transcripts per million reads (TPM) were log2-transformed (*Cushion et al., 2018*).

## Plant material, RNA extraction and qRT-PCR analysis

Normal-grown 3-month-old live *P. edulis* seedlings were used as the control group with the following abiotic stress treatments: 4 °C, 42 °C and 500 ml 30% PEG6000 (The effect obtained by PEG6000 induced water shortage is the same as that obtained by progressive drought of the soil); sampled at 0, 3, 6, 12 and 24 h for the above treatments, and at 0, 3, 6 and 12 h for 42 °C-treated live *P. edulis* seedlings, and the second youngest leaf from top to bottom was snap-frozen in liquid nitrogen and saved in a −80 °C freezer.

Extraction of total RNA using an RNA extraction kit (Kangwei Century Biotechnology Co., Ltd., Taizhou, Jiangsu, China). cDNA was synthesized using a Script RT kit (Takara Bio, Kusatsu, Japan) kit and used for subsequent qRT-PCR assays. For the 11 identified *PeCAMTA* genes, qRT-PCR primers were designed online using Primer Premier 3, with *P. edulis* NTB (nucleotide tract-binding protein) as the internal reference gene (*Fan et al., 2013*). SYBR qPCR Master Mix (Code. Q311-02, Nanjing, China) was used to perform qRT-PCR in Multiplate™ 96-well PCR plates (Bio-Rad, Hercules, CA, USA). Each sample was tested using three technical replicates to ensure the accuracy of results. The reaction conditions refer to the method of Ma R (*Ma et al., 2021*).

Statistical analysis was conducted with an analysis of variance (ANOVA) and Independent-Sample T Test using Statistical Package for the Social Sciences (SPSS) software (version 24.0; IBM SPSS Statistics, Armonk, NY, USA).
**Table 1 Physicochemical properties of *PeCAMTAs* genes.**

| ID | Gene name | Number of amino acids | Molecular weight (kDa) | Theoretical pI | Aliphatic index | Grand average of hydropathicity (GRAVY) | Signal peptide |
|---|---|---|---|---|---|---|---|
| PH02Gene18220.t1 | *PeCAMTA01* | 1,024 | 113.88 | 5.74 | 77.09 | −0.456 | NO |
| PH02Gene42704.t1 | *PeCAMTA02* | 925 | 103.43 | 8.2 | 76.1 | −0.477 | N |
| PH02Gene37813.t1 | *PeCAMTA03* | 1,027 | 114.18 | 5.51 | 74.14 | −0.503 | N |
| PH02Gene40726.t1 | *PeCAMTA04* | 1,030 | 114.81 | 5.49 | 75.5 | −0.507 | N |
| PH02Gene36566.t1 | *PeCAMTA05* | 816 | 90.10 | 5.18 | 74.73 | −0.45 | N |
| PH02Gene07259.t1 | *PeCAMTA06* | 851 | 96.14 | 7.61 | 80.82 | −0.47 | N |
| PH02Gene08544.t1 | *PeCAMTA07* | 1,028 | 114.92 | 5.69 | 77.72 | −0.48 | N |
| PH02Gene05448.t1 | *PeCAMTA08* | 1,025 | 114.11 | 5.92 | 76.92 | −0.50 | N |
| PH02Gene05785.t1 | *PeCAMTA09* | 851 | 96.18 | 6.51 | 77.39 | −0.55 | N |
| PH02Gene15267.t1 | *PeCAMTA10* | 1,026 | 114.85 | 5.78 | 75.30 | −0.51 | N |
| PH02Gene16049.t4 | *PeCAMTA11* | 1,031 | 114.95 | 5.88 | 74.03 | −0.567 | N |

# RESULTS

## Identification and characterization of *PeCAMTA* genes in *P. edulis*

Candidate family members were searched by the plant *CAMTA* Pfam (PF04770) model, and a significant E value of no more than $1 \times 10^{-20}$ was set for preliminary screening. A total of 11 *CAMTA* gene family members were obtained by combining gene structure, chromosomal localization, conserved structural domains and other characteristics, and removing gene duplicate transcripts and non-full-length amino acid sequences. As shown in Table 1, the *CAMTA* gene family genes were renamed *PeCAMTA01* to *PeCAMTA11* based on the chromosomal positioning information of the genes. bioinformatics analysis of the protein sequences of the 11 family members showed that the largest protein molecular weight of the *CAMTA* gene family members was 114.92 kD, and the smallest protein molecular weight was 90.10 kD. The amino acid sequence lengths ranged from 816 to 1,031 aa. The isoelectric points lie between 5.18 and 8.2. Two of the family proteins are basic (theoretical isoelectric point > 7) and nine are acidic (theoretical isoelectric point < 7). The aliphatic amino acid index revealed that the thermal stability of the proteins of this family was between 74.03 and 80.82, suggesting that the proteins of this family have small differences in thermal stability. Signal peptide analysis showed that none of the 11 members had signal peptides, indicating that the protein sequences of the *CAMTA* genes of *P. edulis* do not have transmembrane structures.

## Phylogenetic analysis

Phylogenetic analysis of amino acid sequences of *P. edulis CAMTA*, *O. sativa CAMTA*, *Arabidopsis CAMTA*, *Z. mays CAMTA* and *B. distachyon CAMTA* was performed, and we classified the amino acid sequences of *PeCAMTAs* into five subbranches (I~V) (Fig. 1) according to *Wang et al. (2022)* classification method, among which the protein sequences of *Arabidopsis CAMTA* genes were classified into one subclade, and the amino acid sequences of *O. sativa*, *Z. mays* and *P. edulis CAMTA* genes were grouped into one

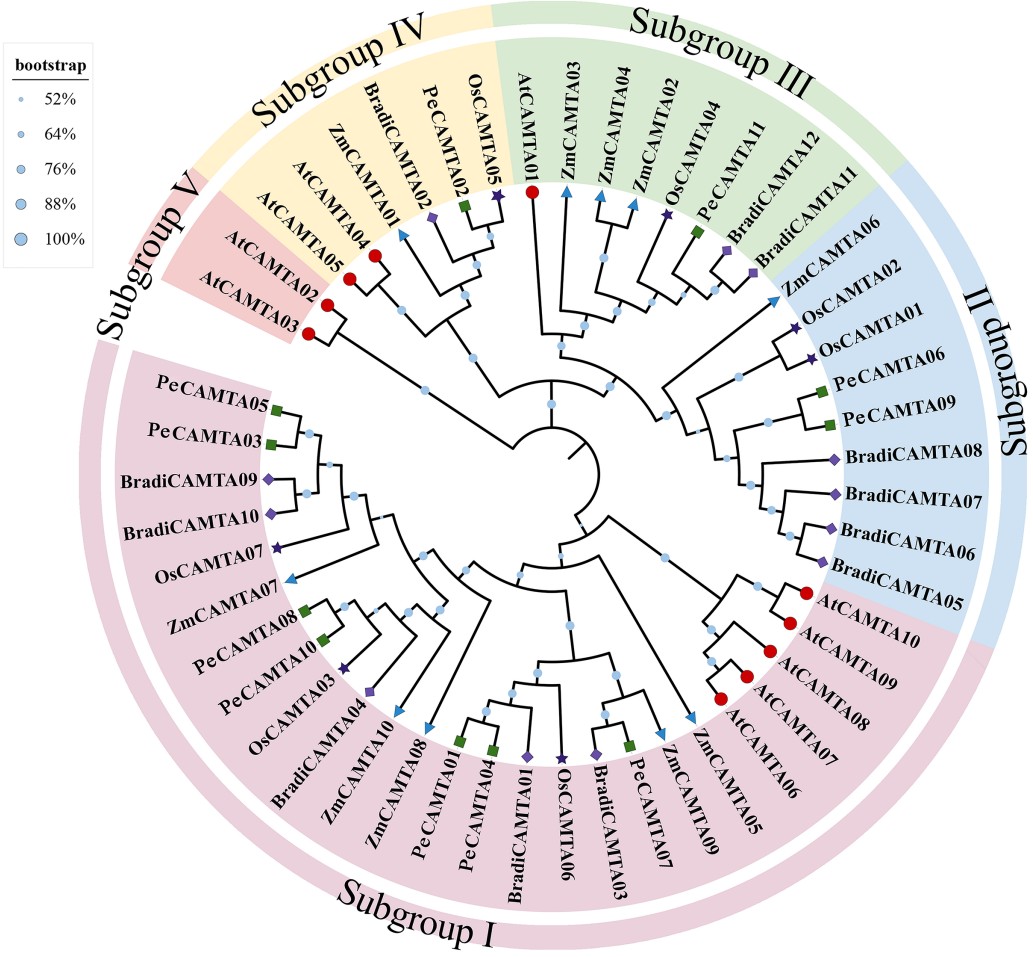

**Figure 1 Phylogenetic analysis.** The full-length amino acid sequences of 50 *CAMTA* proteins were used to construct the phylogenetic tree using MEGA7.0 with the neighbor-joining (NJ) method. The size of graphics at the branch represents the confidence relative value obtained by 100 bootstrap tests. *AtCAMTA* represents *CAMTA* protein sequence of *Arabidopsis thaliana*, *OsCAMTA* represents *CAMTA* protein sequence of rice, *ZmCAMTA* represents *CAMTA* protein sequence of *Zea mays* and *Bradi-CAMTA* represents *CAMTA* protein sequence of *Brachypodium distachyon*.

subclade. It is more closely related to *O. sativa* and *Z. mays*, and more distantly related to *Arabidopsis*.

## Gene structure, conserved domains, motifs and sequence analysis

Analysis of the gene structure of *PeCAMTA* gene family showed that the number of introns (intron) of each *PeCAMTA* gene ranged from 10 to 14. The 11 sequences were divided into four categories, because the affinities of *P. edulis* in other species make the results differ from the classification in the evolutionary tree. Gene *PeCAMTA09* in subfamily III contains the longest intron region, while gene *PeCAMTA04* in subfamily II and gene *PeCAMTA11* in subfamily III have the shortest introns.

*PeCAMTA* gene family was further analyzed for conserved structural domains based on the NCBI online software CDD, as shown in Fig. 2C. As shown, all *CAMTA* gene family
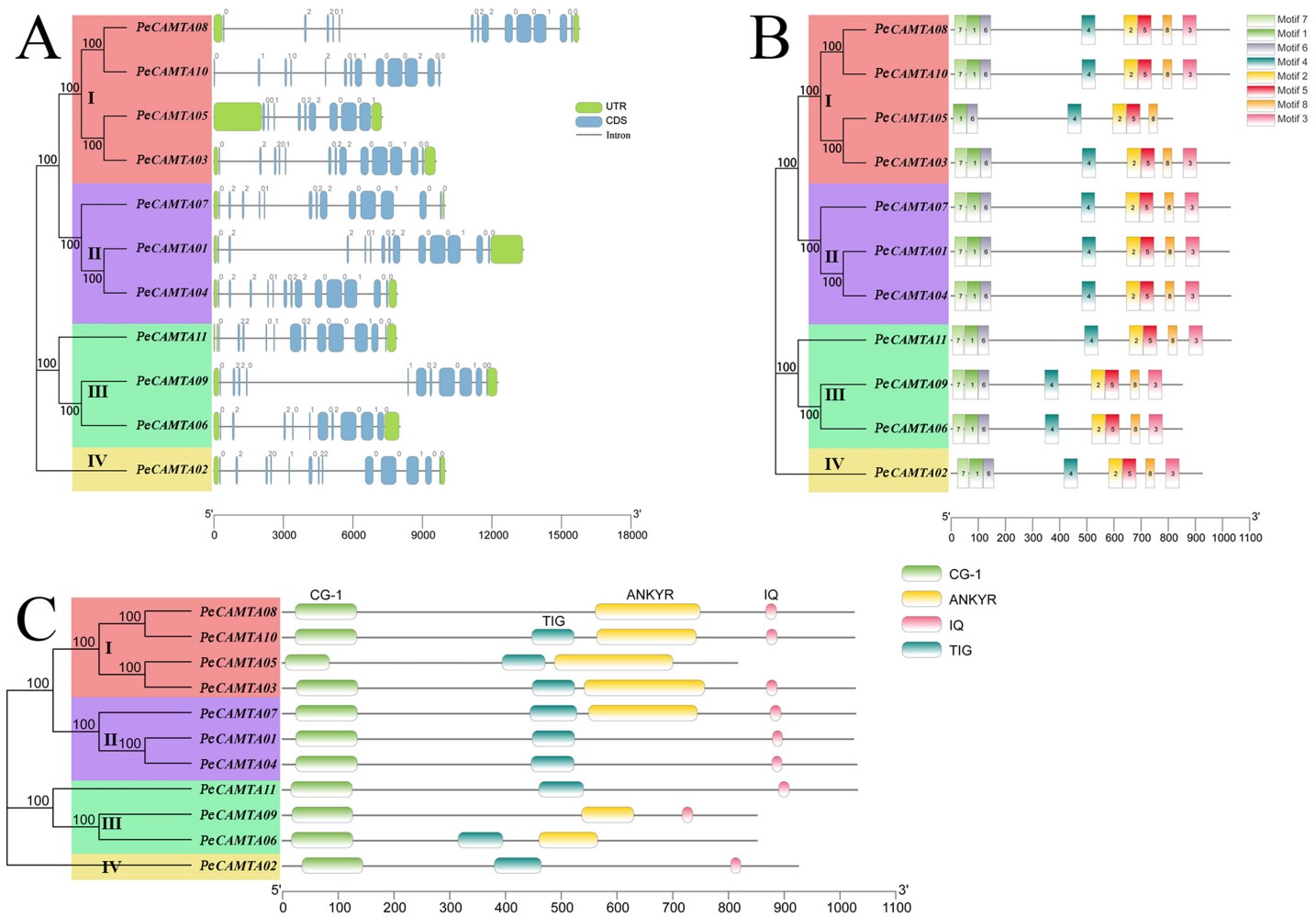

**Figure 2 Gene structure, conserved motifs and conserved domains of *PeCAMTAs*.** Phylogenetic trees were made with maximum likelihood by using the Neighbor joining model and MEGA 7.0 software. Different colors plates represent different groups. (A) Exon–intron distribution of *PeCAMTAs*. (B) Conserved motifs in *PeCAMTAs*. Motif 1 to motif 8 represented different motifs, and they were represented by different color boxes on the right. (C) Conserved domains in *PeCAMTAs*. CG-1, CG-1 domains. TIG, IPT/TIG domain. ANKYR, ankyrin repeats; IQ, is a calmodulin-binding motif.

members contained CG-1 structural domains located at the N terminus, *PeCAMTA10*, *PeCAMTA05*, *PeCAMTA03* in the first subclade and *PeCAMTA11*, *PeCAMTA06* in the third subclade had TIG structural domains in addition to the typical CG-1 structural domains, while *PeCAMTA08* in the first subgroup and *PeCAMTA09* in the third subgroup do not have TIG structural domains, and both the second and fourth subgroups contain both CG-1 and TIG structural domains. All of the first subgroup contained ANKYR structural domains, *PeCAMTA07* in the second subgroup and *PeCAMTA09* and *PeCAMTA06* in the third subgroup contained ANKYR structural domains, and the rest of *PeCAMTA01* and *PeCAMTA04* in the second subgroup, *PeCAMTA11* in the third subgroup and the fourth subgroup did not contain ANKYR structural domains. All of the second subclade contained IQ structural domains, *PeCAMTA08*, *PeCAMTA10*, *PeCAMTA03* in the first subclade, *PeCAMTA11*, *PeCAMTA09* in the third subclade and
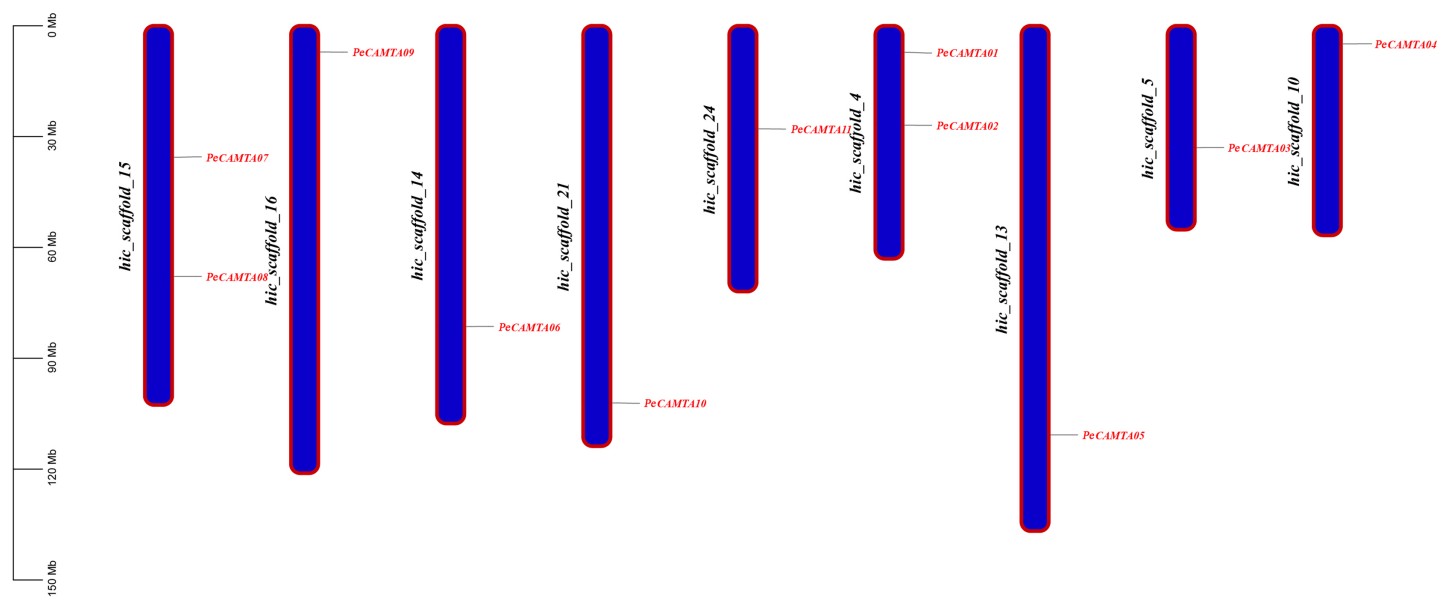

**Figure 3 The distribution and duplication events of *PeCAMTAs* on the chromosome.** The location of these genes on the chromosome was visualized using the visualization tools.

the fourth subclade contained IQ structural domains, and the remaining *PeCAMTA05* in the first subclade and *PeCAMTA06* in the third subclade did not have IQ structural domains. *PeCAMTA08*, *PeCAMTA10*, *PeCAMTA03* in the first subfamily and *PeCAMTA07* in the second subfamily have all *CAMTA* structural domains.

The members of the *P. edulis CAMTA* gene subgroup contain motifs numbering 6 and 8, which are highly conserved, of which motif 1, motif 6 and motif 7 constitute the CG-1 structural domain. Except for *PeCAMTA05*, which lacks motif 7 and motif 3 in the first family, the genes in the other families have all motifs.

## Chromosomal location and gene duplication of *PeCAMTA* genes

The chromosome distribution (Fig. 3) of the *PeCAMTA* gene family showed that 11 *PeCAMTA* genes were distributed on nine chromosomes with different densities of chromosomal gene distribution. Based on intraspecific covariance analysis (Fig. 4A), only the genes *PeCAMTA08* and *PeCAMTA10* were found to have undergone gene duplication (tandem duplication), while the remaining genes did not show gene duplication. The results indicated that only individual genes caused amplification of *CAMTA* transcription factor members on different chromosomes through gene duplication.

As shown in Fig. 4, only three *P. edulis CAMTA* homologous protein genes occur in *Arabidopsis* chromosomes, while eight *P. edulis CAMTA* genes can be found on six *Z. mays* chromosomes with corresponding paralogous homologs, and nine *P. edulis CAMTA* genes can be found on five *O. sativa* chromosomes with corresponding paralogous the same genes were found on five *O. sativa* chromosomes. Therefore, the covariance between *P. edulis* and *O. sativa* and *Z. mays* was more significant than that between *P. edulis* and *Arabidopsis*. In addition, most of the genes in the *O. sativa* and *Z. mays CAMTA* families

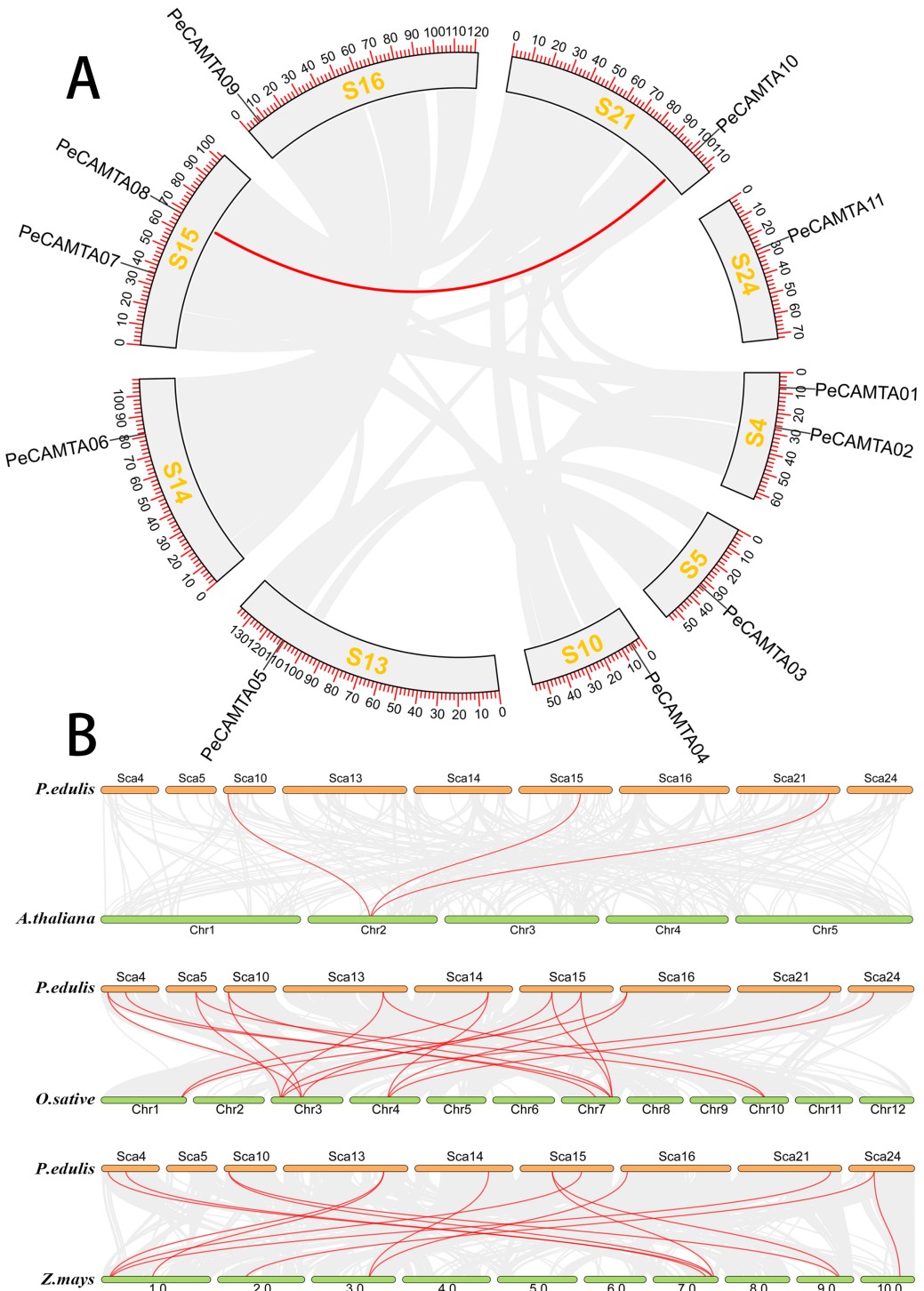

**Figure 4 Colinearity analysis.** (A) Intraspecific colinearity analysis. A total of 11 *PeCAMTAs* were mapped onto the chromosomes on the basis of their physical location. Chromosome numbers (scaffold1–scaffold24) are distributed in the outer circle, the red lines indicate duplicated *PeCAMTA* gene pairs. (B) Analysis of collinearity between different species. The gray lines indicate duplicated blocks, while the red lines indicate duplicated *PeCAMTA* gene pairs. Chromosome numbers are at the bottom of each chromosome.

have more than two paralogous homologs in *P. edulis*, inferring that there may have been a massive gene doubling event in the *P. edulis* CAMTA gene family in the evolution process.

## Cis-element analysis of *PeCAMTAs*

The *P. edulis CAMTA* gene family members contain 11 genes extracted upstream to 1,500 bp nucleotide sequences, and promoter prediction revealed that in addition to the core promoter elements, many other cis-acting elements were found (Fig. 5), such as light-responsive elements, hormone-response-related elements and stress-responsive elements related to plant growth and development. The most abundant were hormone response-related elements, with all gene promoters containing at least one light response element and most gene promoters containing at least one phytohormone response element. The stress response elements include low temperature stress response components, drought stress response components, anaerobic induction response components and other abiotic stress response components. All *PeCAMTAs* contained the drought stress response component MYC, and the drought stress response component MYB was the most abundant response element, suggesting that *PeCAMTAs* plays an essential role in drought stress response. The results suggest that different components of the promoter region of *P. edulis CAMTA* gene family may be important in regulating plant growth and development and in resisting abiotic stresses.

## Tissue-specific expression levels of *PeCAMTA* genes

To study expression of *CAMTAs* family genes (transcriptome) in different organs of *P. edulis* (Fig. 6). The expression levels of *CAMTAs* in four tissues (leaf, stem, whip and root) were assessed by RNA-seq data. Gene expression profiles in different tissues indicated that *CAMTA* has different functions in *P. edulis*. The results showed that *PeCAMTA07* and *PeCAMTA11* expression profile was higher than other genes. Except for *PeCAMTA11*, the expression of the other *PeCAMTA* genes was higher in leaves than in stems, whips and roots. Moreover, *PeCAMTA07* was more highly expressed in each tissue, indicating that this gene plays an important role in the overall development of *P. edulis*.

## Expression profiles of the *PeCAMTA* genes during abiotic stress

To investigate the expression of *PeCAMTAs* during abiotic stress, we analyzed the expression of 11 *PeCAMTAs* under three abiotic stresses using qRT-PCR: polyethylene glycol (PEG), heat, and cold treatment. The expression patterns of *PeCAMTAs* responded differently to the three abiotic stresses, and some *PeCAMTAs* were either significantly induced or repressed. The expression pattern of most genes changed significantly during the early phase (0–6 h) of the stress response.

Under cold stress (Fig. 7A), expression peaked at 6 h for all genes except *PeCAMTA02*, with a trend of up-regulation followed by down-regulation, with *PeCAMTA02* peaking at 3 h followed by down-regulation. *PeCAMTA03* and *PeCAMTA06* expression peaked higher than the other genes, at about 6. *PeCAMTA01*, *PeCAMTA02*, *PeCAMTA03*, *PeCAMTA05*, *PeCAMTA07*, *PeCAMTA08*, *PeCAMTA10* and *PeCAMTA11* expression was below about 0.3 at 12 h and tended to be 0 at 24 h. *PeCAMTA04*, *PeCAMTA06*,
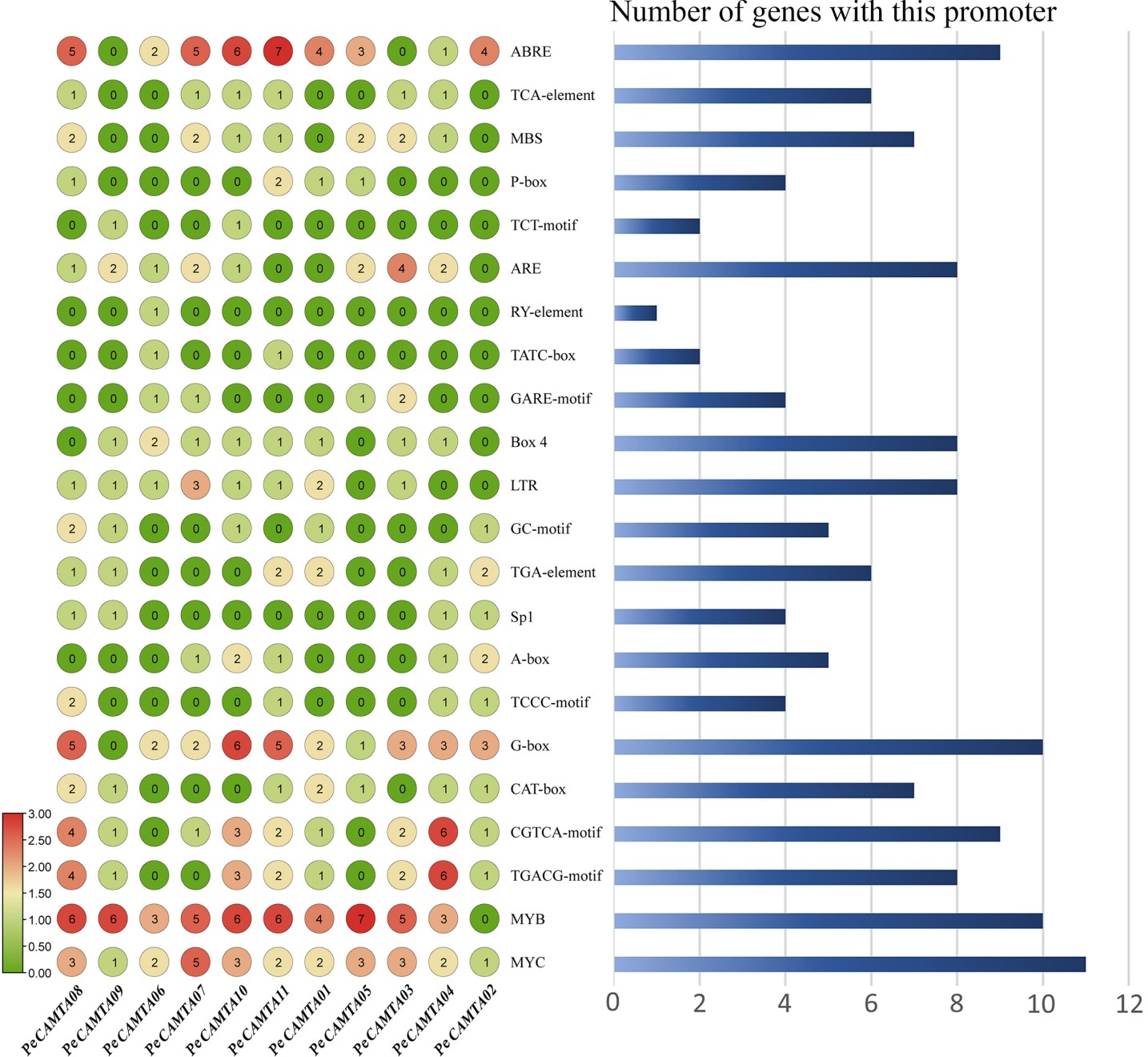

**Figure 5 Cis-element analysis of *PeCAMTAs*.** The graph on the left shows the enrichment of cis-acting elements and the numbers in the circles represent the number of cis-acting elements. The graph on the right shows a proportional map and the number indicates the number of genes containing that cis-acting element.

*PeCAMTA08, PeCAMTA09* and *PeCAMTA10* expression at 3 h was significantly lower than in the untreated condition (CK).

Under heat stress (Fig. 7B), all genes reached peak expression at 6 h, with higher peaks at 22.4 for *PeCAMTA03* and 24 for *PeCAMTA11*, and the expression trends were all up-regulated and then down-regulated. All genes were expressed at lower levels at 3 h than

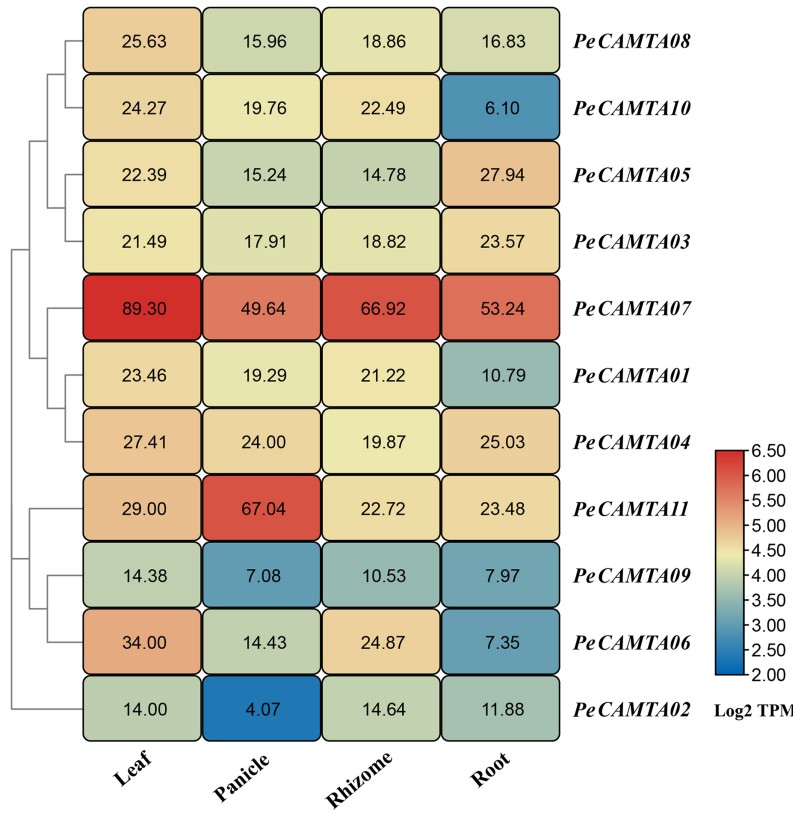

**Figure 6 Expression profile cluster analysis of *PeCAMTAs* with differential tissue expression.** Heatmap showing relative expression levels of *PeCAMTAs* in roots, leaves, panicles, and rhizomes. The values are expressed with log2 TPM.

untreated (CK), with *PeCAMTA04*, *PeCAMTA06*, *PeCAMTA07*, *PeCAMTA08*, *PeCAMTA09* and *PeCAMTA11* all converging to 0 at 3 h.

Under drought stress (Fig. 7C), all genes reached peak expression at 6 h, with the highest peak being 34 of *PeCAMTA11*. The expression trend was mostly up-regulated, then down-regulated and finally up-regulated, with only *PeCAMTA06* and *PeCAMTA10* having lower expression at 24 h than at 12 h. All genes except *PeCAMTA01* had no significant difference in expression from the control (CK) at 3 h.

To better compare the differences in expression of the *PeCAMTA* gene family under the three stresses, the expression values of each gene under different stresses were compared together (Fig. 8). The comparison showed that, except for a few cases (*e.g.*, *PeCAMTA11*, where heat stress expression was higher than drought stress at 12 h), the majority of cases were higher for drought stress than for heat and cold stress, whether at 3, 6, 12 or 24 h. In addition, drought stress expression was up-regulated at 24 h.

To compare expression differences between different genes, the expression of all genes under the same stress was compared (Fig. 8). Under cold stress, *PeCAMTA11* had higher expression at 3 h and *PeCAMTA03* at 6 h, while *PeCAMTA09* had higher expression at both 3 and 6 h. Under drought stress, *PeCAMTA08* had lower expression at all stages, *PeCAMTA11* had higher expression at 6 h and was in an average position at both 12 and

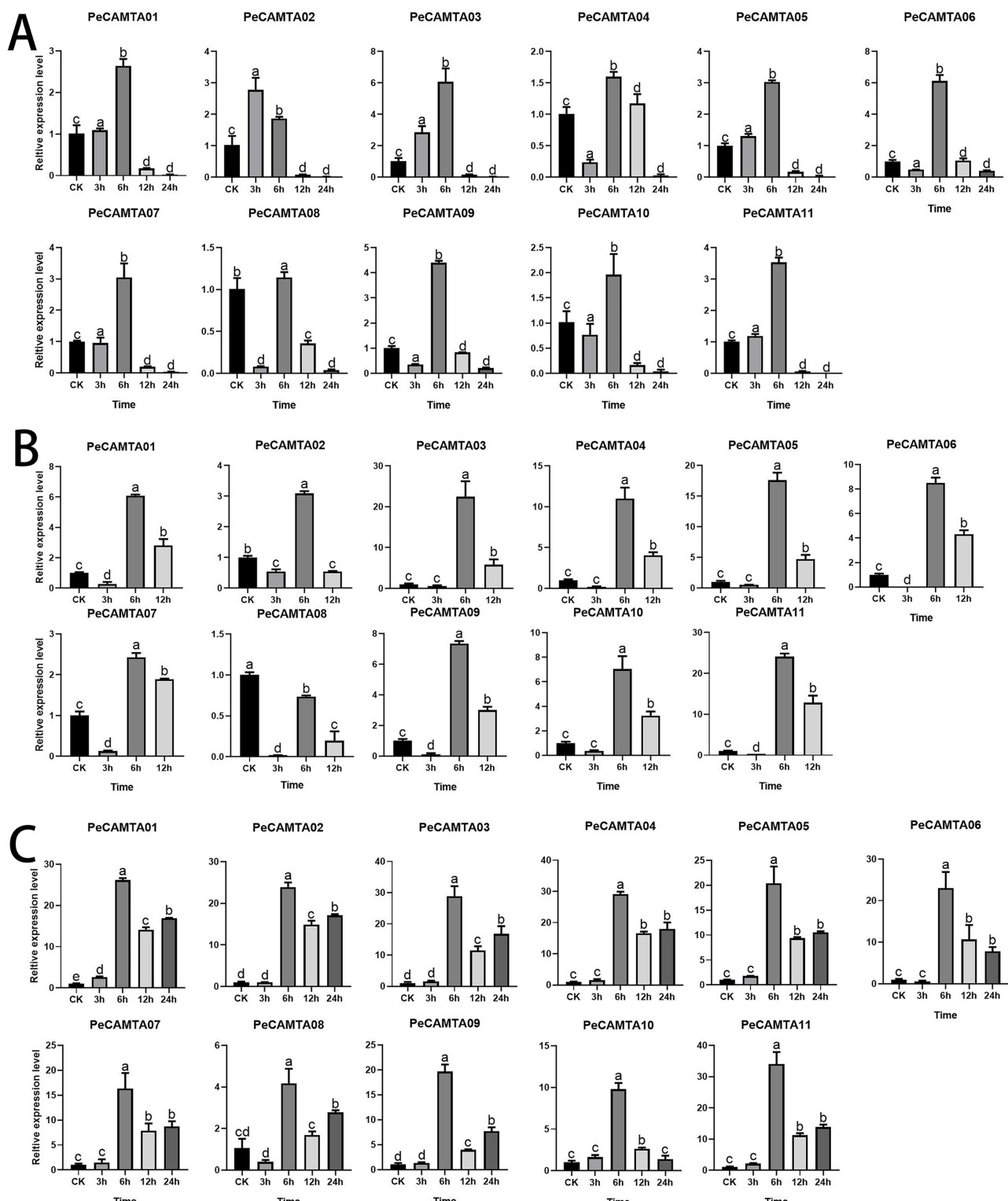

**Figure 7** **Expression analysis of 11 selected *PeCAMTAs* genes under abiotic stress treatment using qRT-PCR.** (A) Cold stress treatment; (B) heat stress treatment; (C) drought stress treatment. Bars with same letter means no significant difference based on LSD test ($p \leq 0.05$).

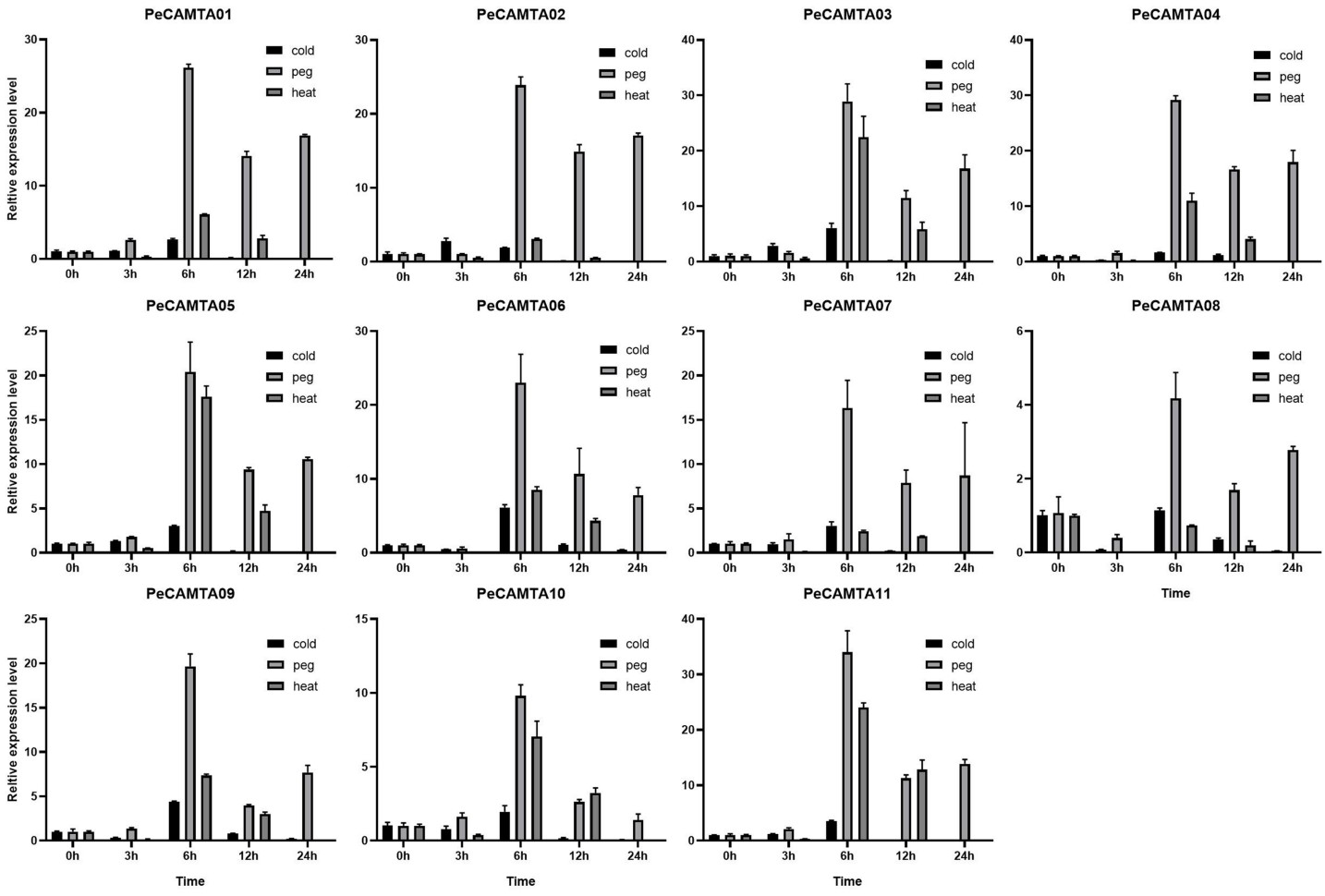

**Figure 8 Comparison of qRT-PCR results for three abiotic stresses in the same *PeCAMTAs* gene.** Cold stress treatment; heat stress treatment; drought stress treatment.

24 h. Under heat stress, *PeCAMTA03* and *PeCAMTA11* had higher expression at 6 h and *PeCAMTA11* also had higher expression at 12 h.

## DISCUSSION

### Genome-wide identification and phylogenetic analysis of the *PeCAMTA* gene family

*CAMTAs* are a specific class of plant transcription factors that play an essential role in the regulation of plant growth and development and metabolism (*Galon et al., 2008*; *Yang et al., 2012*; *Yang & Poovaiah, 2002*). The molecular functions of *CAMTA* have been verified not only in *Arabidopsis* (*Galon et al., 2010*; *Pandey et al., 2013*) and *O. sativa* (*Gain et al., 2022*) as model plants, but also in cotton (*Pant et al., 2018*), maize (*Yue et al., 2015*), tobacco (*Kakar et al., 2018*) and tomato (*Yang et al., 2012*), where the *CAMTA* gene family has been gradually identified. However, no studies on *CAMTAs* have been conducted in the economically important bamboo species, *P. edulis*. Currently, the draft genome of *P. edulis* is largely complete, allowing for a full identification of key gene families (*Peng*

*et al., 2013*; *Zhao et al., 2018*). We classified the 11 *PeCAMTA* genes into five categories, Class I, Class II, Class III and Class IV, based on phylogenetic analysis. Among them, four members (36%) belonged to Class I, three members (27%) to Class II, three members (27%) to Class III, and one member (9%) to Class IV (Fig. 1).

Gene structure analysis reveals structural differences among members within the same subfamily. Such as, *PeCAMTAs* members in the same family I have intron numbers ranging from 10–14. Therefore, we hypothesize that members of subfamily I may have undergone pruning of gene fragments during their evolution (*Li et al., 2016*; *Staiger & Brown, 2013*). Nevertheless, the similar conserved sequences and gene structures among *CAMTA* gene family members suggest that gene biological functions are generally the same within a family. All six NTR1 homologs of *Arabidopsis* have a conserved structural feature with a DNA-binding region (CGCG structural domain) at the N-terminal end and a CaM-binding structural domain at the C-terminal end. The role of $Ca^{2+}/CaM$ may be expressed in controlling interactions with other proteins or altering transcriptional activation of other proteins. In addition, conserved domain comparison showed that all *PeCAMTA* genes have CG-1 structural domains, indicating that the conserved motifs of the *CAMTAs* family are broadly conserved during evolution.

During signal transduction, multiple cis-acting elements on a gene promoter work together to regulate multiple complex biological responses. *Solanum lycopersicum CAMTA* gene contains salt stress regulatory elements, including ABRE, G-box, MBS, and TGA (*Wang et al., 2021*), and *CAMTAs* of different species have been reported to respond to a variety of biotic and abiotic stresses, including low temperature, hormones, high salt, and drought. Two genes, *ZmCAMTA4* and *ZmCAMTA6*, were highly expressed under stress treatment, and cis-element analysis revealed the involvement of *CAMTA* genes in the association between environmental stress and stress-related hormones, and the *GhCAMTA* gene family may also be involved in the phytohormone signaling pathway (*Liu, Hewei & Min, 2021*; *Pant et al., 2018*). On the basis of the PlantCARE software, we found that elements involved in abscisic acid response, MeJA response, growth hormone (IAA) and many other hormone regulation-related elements were present. Therefore, we suggest that *PeCAMTA* genes may also be involved in the stress response of plants. Interestingly, the promoter regions of most *PeCAMTA* genes have the largest number of MYB elements involved in drought induction (Fig. 5). Recent studies on wheat confirmed that the expression of *TaCAMTA1a-B* and *TaCAMTA1b-B*. A total of 1 was down- and up-regulated, respectively, in response to drought stress to maintain normal physiological functions associated with the plant, and wheat *CAMTA* gene family members also contain a large number of MYB elements (*Wang et al., 2022*). The previous study also gave us ideas for subsequent stress experiments, and it became important to verify the expression of *PeCAMTAs* in response to drought stress.

## Evolutionary characterization of the *PeCAMTA* gene family

Gene duplication may produce new genes, which greatly helps in the evolution of gene function. The three evolutionary patterns of gene replication are (*Liu et al., 2019*): segmental duplication, tandem duplication and translocation events. Segmental and

tandem replication are the most common basis for gene family expansion in plants (*Freeling, 2009*; *Li & Barker, 2020*). Previous studies on whole genome duplication have shown that the genome size of bamboo (2,051.7 Mb) is similar to that of its close relative *Z. mays* (2,066.4 Mb), but the number of *CAMTA* genes is higher in bamboo than in maize (*Chen et al., 2020*). Therefore, we performed a consistency analysis within and among the *P. edulis* genomes. Within the *P. edulis* genome, there was one pair of segmental duplication genes in the *CAMTA* gene. Therefore, the amplification of the *CAMTA* gene family mainly comes from gene fragment replication. Simultaneous analysis of the genome of Bamboo and three other sequenced plant genomes showed that the members of the bamboo *CAMTA* gene family had significant consistency with the genomes of the monocot plant *O. sativa*.

## The role of *PeCAMTA* genes in different tissues and organs

Several studies have shown that *CAMTAs* can regulate plants during lateral organ development and have an important effect on plant organ formation (*Rahman et al., 2016*; *Shangguan et al., 2014*; *Wang et al., 2015*; *Yang et al., 2015*), which is consistent with our findings. Analysis of expression profiles in different tissues of bamboo revealed that a large number of *PeCAMTAs* showed the amount of expression varies in different tissues (Fig. 6). *PeCAMTA07* was up-regulated in all *P. edulis* growth sites, with expression in leaves, whips and roots more than twice that of other genes, indicating its importance in *P. edulis* growth and development. *PeCAMTA11* was most abundantly expressed in flowers, suggesting that *PeCAMTA11* is involved in flower bud differentiation in *P. edulis*, but other *PeCAMTAs* were more abundantly expressed in leaves than in flowers, suggesting that they are mainly involved in the plant growth process, but not in the plant bud differentiation process.

## Expression of *PeCAMTA* genes in responses to cold, drought and heat treatments

Studies have shown that regulatory elements in the promoter region are important for the regulation of the gene in different environments (*Wang et al., 2023*). The $E_2$F/DP gene in *P. edulis* is mainly involved in drought stress response, and the Phe$E_2$F/DP promoter contains many MYB and MYC2 binding sites (*Li et al., 2021*). Many MYB and MYC2 promoters were also identified in the *TaCAMTA* gene family, and *TaCAMTA* plays an important role in the physiological activities of wheat in response to drought stress. Using real-time fluorescence quantification assays, the results of the *PeCAMTAs* promoter analysis were combined to verify whether *PeCAMTAs* is involved in drought stress regulation. Comparison with other stresses showed that the expression of the *PeCAMTA* gene family was significantly higher under drought stress than under heat and cold stress, with higher expression at 6, 12 and 24 h. The expression of *PeCAMTA11* was superior under all three stresses, which could be used as a reserve for the next gene function validation assay. Our study suggests that the *PeCAMTA* gene family plays a crucial role in drought stress response, but further studies are needed to elucidate the functional significance of the *CAMTA* gene family in *P. edulis*.

## CONCLUSIONS

Our results present new findings for the *P. edulis CAMTA* gene family and provide partial experimental evidence for further validation of the function of *PeCAMTAs*.

## ACKNOWLEDGEMENTS

We are thankful to Bin Huang and all the students and teachers who have helped us.

### Funding

The authors received no funding for this work.

### Competing Interests

The authors declare that they have no competing interests.

### Author Contributions

- Ce Liu conceived and designed the experiments, performed the experiments, analyzed the data, prepared figures and/or tables, authored or reviewed drafts of the article, and approved the final draft.
- Dingqin Tang conceived and designed the experiments, authored or reviewed drafts of the article, and approved the final draft.

### Data Availability

Raw data, gene renaming, quantitative primer design and covariance source files are available in the Supplemental Files.

### Supplemental Information

Supplemental information for this article can be found online at http://dx.doi.org/10.7717/peerj.15358#supplemental-information.

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
