# Peer review of "Comprehensive identification and expression analysis of CAMTA gene family in Phyllostachys edulis under abiotic stress"

_PeerJ, doi:10.7717/peerj.15358_

## Round 0.1 · original submission · Major Revisions

Dear Author

The reviewers have recommended revisions to your manuscript. Therefore, I invite you to respond to the reviewer's comments and revise your manuscript.
In addition, there are significant concerns about the manuscript's grammar, usage, and overall readability. We, therefore, request that you revise the text to fix the grammatical errors and improve the overall readability of the text.

With Thanks

·

Basic reporting

Title: Comprehensive identiûcation and expression analysis of CAMTA gene family in Phyllostachys edulis under abiotic stress.
The manuscript has presented numerous bioinformatics study but it needs a lot of improvement in writing as well as data representation.
1. Authors have not described the term CAMTA in the manuscript. Full form of CAMTA should come in the background, abstract and introduction when it appears first time.
2. Some typological errors like CAMATA, dentification, ChroMosome etc. should be corrected.
3. Several statements are not clear and some of them are mentioned below:
a) “CAMATA was discovered when part of the ……………(Iqbal et al. 2020)”.
b) “The Sequence Toolkits module of TBtools software (v1.098765)……….”.
c) “The expression of Moso bamboo CAMTA was higher…………………………”.
d) “For example, PHCAMTA07 was highly expressed………………………………..”.
e) “The whole genome information…………………………..Brachypodium distachyon”.
f) “Transcriptome data, quantified as transcripts per…………………………”.
g) “The PheE2F/DP promoter in response to drought………………………….”.
h) “Several studies have shown that CAMTAs can regulate plants………”.
4. What is “PHCAMTA”? Describe it at the first time. Why it is not ‘Ph’ in spite of ‘PH’?
5. All the gene names as well as scientific names should be italicized throughout the manuscript.
6. The authors have mentioned about genome wide identification of rice CAMTA but recent references on that is not mentioned anywhere.
7. Some statements are not complete eg. “but the number of CAMTA families is higher than that of the latter”.
8. Why different styles have been followed for different subheadings? Rectify it.
9. Discussion part is very poorly written. Under the subsection “Expression of PHCAMTA Genes in Responses to Cold, Drought and Heat Treatments” only drought stress have been mentioned without any meaningful approach.
10. This study identified 11 unique CAMTAs from Phyllostachys edulis and authors stated that “Signal peptide analysis showed that none of the 23 members…..”. What are these 23 members?
11. “In reference to (Dezhou Wang) (Wang et al. 2022), the….” is not a good start in results.
12. Justify or rewrite the statement “The analysis revealed that the amino acid sequences of PHCAMTA…..”.
13. Is Moso bamboo CAMTA and PHCAMTA are same? If yes, check “ChroMosomal location and gene duplication of PHCAMTA genes”.
14. Check and re-write the results and discussion part. Discussion part should include proper references in relation to the original findings.
15. Conclusion should be precise and specific. It is written in a elusive manner.
16. In reference part also different formatting is evident (eg. Line 523 and 526).

Experimental design

It looks fine

Validity of the findings

Findings are valid but representation is faulty and it might create difficulty among the readers or scientific community.

Additional comments

The manuscript has presented numerous bioinformatics study but it needs a lot of improvement in writing as well as data representation. Modify Figure 5 and 6 in scientific manner. Figure legends are too small and a lot of informations are missing.

Reviewer 2 ·

Basic reporting

The authors identified calmodulin-binding transcriptional activators (CAMTAs) in moso bamboo (Phyllostachys edulis) by using genome data from available databases. They performed bioinformatic analysis such as: physicochemical property analysis, signal peptide analysis, phylogenetic tree construction, gene structure-conserved domain-motif analysis, cis-acting element analysis, chromosome distribution and covariance analysis, tissue specific expression analysis from RNA-seq data. They also performed experimental analysis for the abiotic stress response of CAMTA genes with qRT-PCR in Phyllostachys edulis.
Although I think this study could provide contribution to understand the role of CAMTA genes under stress conditions in plants, I believe that the manuscript, as it stands, does not meet the criteria for publication in PeerJ.
There are lots of English grammar mistakes throughout the manuscript. This manuscript should be checked by a fluent English speaker after all corrections are made.
There are half sentences, and commands and lots of sentences hard to understand in the manuscript. The manuscript should be rewritten carefully.
Authors knows that CAMTAs are first identified in tobacco, however it is a great shortcoming not to reference the article to which they were first identified.
Giving a species name (Phyllostachys edulis) and calling it a “genus” is a big mistake in biological knowledge. Additionally, the authors paid no attention to the italicization of species and genus names throughout the manuscript.
In materials and methods section, authors talk about a different species: P. chinensis, not P. edulis. Moreover, they talk about Brachypodium distachyon in materials and methods, but they include “pepper” in results and figures.
I believe that the manuscript needs major revision for publication in PeerJ.

Experimental design

no comment

Validity of the findings

no comment

Additional comments

Abstract
In “Background” section:
"Arabidopsis thaliana" should be italicized. Species names should be italicized and should be checked throughout the manuscript.
In “Results” section:
“CAMTA gene was” should be “CAMTA genes were”
“Abiotic stress on moso bamboo was also found to be involved in drought stress response” should be rewritten to be meaningful
In “Conclusions” section:
“CAMTA family” should be “CAMTA gene family”
“PHCAMTAs” should be “PeCAMTAs” if that refers to the Phyllostachys edulis CAMTA genes and should be corrected throughout the manuscript.

Introduction
Line 40: “thus mediating various stress responses in plants” is grammatically incorrect after the first sentence and should be rewritten separately from the first sentence.
Line 41: “CaM is a ubiquitous eukaryotic Ca2+ sensor that binds Ca2+ into a flexible Ca2+/CaM structural protein, which, together with the ability of Ca2+ to interact with a number of proteins, allows CaM to regulate protein targets in many different signaling pathways” sentence is too long. Authors should use shorter sentences to be understood better.
Line 46: “in response to plant responses” Authors should avoid using same words.
Line 47: “CAMTA” is first used in the text and should be stated that this is the abbreviation of calmodulin-binding transcriptional activators
Line 47: “CAMTA, a major transcription factor regulated by calmodulin (CaM), was first identified in tobacco in 2009 (Kim et al. 2009).” This is a wrong information. The reference that Authors give is a review. CAMTA proteins were first identified in tobacco in 2000. And the reference for that is:
Yang, T., & Poovaiah, B. W. (2000). An early ethylene up-regulated gene encoding a calmodulin-binding protein involved in plant senescence and death. Journal of Biological Chemistry, 275(49), 38467-38473.
Line 53: Authors gave example to IQ motifs as “(IQXXXRGXXXR)”, however “[I,L,V]QXXXRXXXX[R,K]” is a more generalized IQ motif and clearly indicated in the reference that Authors gave: Bähler & Rhoads 2002. I think Authors should avoid giving a motif here.
Line 54: “CAMATA was discovered when part of the cDNA clone (CG-1) was isolated from parsley and subsequently reported in various multicellular organisms (Iqbal et al. 2020).” is very irrelevant at the end of the paragraph and should be removed completely.
Line 57: “It has been found that CAMTA transcription factors exhibit very important and simple and effective functions in plant growth and development, biotic and abiotic stress (e. g. low temperature stress) responses, and that CAMTAs of different species respond to various biotic and abiotic stresses including low temperature, hormones, high salt and drought to varying degrees (Chung et al. 2020; Noman et al. 2021; Shkolnik et al. 2019; Yue et al. 2015).” Sentence is too long to follow. Are CAMTA TFs very important or simple? Authors should avoid using opposite adjectives in the same sentence. Authors gave “low temperature” abiotic stress example 2 times in a sentence; repeats should be avoided. That sentence should be rewritten and divided into at least 2 sentences.
Line 62: “Brassica napus” and “CAMTA3” should be italicized. Gene names should be italicized throughout the manuscript.
Line 63: The DOI for (Luo et al. 2021) is not working and should be removed from references. “ZmCAMTA4 and ZmCAMTA6” should be italicized.
Line 66: “Ming wei (Wei et al. 2017) suggested that PtCAMTA genes play an essential role in resistance to cold stress, and he showed that woody plants and crops have different CAMTA gene expression patterns under abiotic stresses and phytohormone treatments.” should be written as “Wei et al. (2017) suggested that Populus trichocarpa CAMTA genes play an essential role in resistance to cold stress, and showed that woody plants and crops have different CAMTA gene expression patterns under abiotic stresses and phytohormone treatments.”
Line 69: “Gossypium hirsutum” and “GhCAMTA11, GhCAMTA7, GhCAMTA14” should be italicized.
Line 73: “HbCAMTA3” Authors should give the full name of the species when they first mention about it. Should be as “Hevea brasiliensis CAMTA3”
Line 74: “AtCAMTA3 and Arabidopsis” should be italicized.
Line 76: “Interestingly, it was found that TaCAMTA mainly responds to drought stress in wheat in reaction to various abiotic stresses in the nursery stage, and TaCAMTA1b-B. 1 plays an essential role in the response to drought stress caused by water deficit in the nursery stage (Wang et al. 2022).” What does “responds to drought stress in wheat in reaction to various abiotic stresses” means? Authors should have this manuscript checked for English grammar.
Line 80: “Phyllostachys edulis” is NOT a genus, it is a species name. “Phyllostachys” is a genus name.
Line 84: “We comprehensively analyzed the phylogenetic relationships between moso bamboo and model plants in the CAMTA gene family to elucidate their evolutionary relationships.” that sentence doesn’t represent what authors intend to say. It should be “We comprehensively analyzed the phylogenetic relationships of the CAMTA gene family in moso bamboo and model plants to elucidate their evolutionary relationships.”
Line 86: “PHCAMTA” should be “PeCAMTA” and it should be checked throughout the manuscript
Line 88: What does “stressful abiotic stresses” represents? and “P. edulis” should be italicized.
Line 90: “P. edulis” should be italicized.

Materials and Methods
Line 93: “dentification” should be “Identification”
Line 97: The link “(https://myhits.sib.swiss/cgi-bin)” that authors gave is not working.
Line 110: “Properties, signal peptides were analyzed.” should be “properties and signal peptides were analyzed.”
Line 112: Arabidopsis thaliana? Zea mays? The species names should be given fully and should be italicized. “Brachypodium distachyon” should be italicized.
Line 114: (http://www.arabidopsis.org) database is given for Zea. The correct database should be given.
Line 115: “Brachypodium distachyon” should be italicized. (http://plants.ensembl.org/) is not a specific database for Brachypodium distachyon. Therefore authors should not say “Brachypodium distachyon database”
Line 117: What is “M. spp” ?
Line 127-128: There are no sentences here. There are only commands.
Line 137: Authors indicate that they performed gene expression analysis of P. chinensis. Why did authors make all bioinformatic analysis on Phyllostachys edulis, but gene expression on P. chinensis?
Line 143: The reason of the usage of PEG should be stated clearly.
Line 154: Authors should give statistical analysis method here.

Results
Line 158: “Eleven candidate family members were searched” should be replaced with “Candidate family members were searched” because 11 candidates is your result, not your method.
Line 176: Arabidopsis and Zea should be given with species name and should be italicized.
Line 180: Are P. edulis and PHCAMTA different from each other?
Line 181: Arabidopsis and Zea should be given with species name and should be italicized.
Line 206: “mauve CAMTA gene family” should be replaced by “mauve CAMTA gene subgroup” and
if authors mention “mauve CAMTA genes”, they also should cite the related figure.
Line 212: If authors could detect on which chromosomes did 11 CAMTA genes were distributed, why did not they name them as chromosomes? Why did authors name them as scaffolds? A scaffold is a portion of the genome sequence reconstructed from end-sequenced whole-genome shotgun clones. A scaffold consists of contigs and gaps and does not mean a chromosome.
Line 213: Authors say PHCAMTA08 and PHCAMTA10 underwent gene doubling. In Fig3, CAMTA08 seems to be in the same scaffold with CAMTA07. However, CAMTA10 seems alone on a scaffold. In contrast, CAMTA01 and CAMTA02 seems to be in the same scaffold.
Line 217: To which organism do authors refer by “pepper”? Brachypodium distachyon is not pepper. If they refer to “Capsicum annuum”, then why did not authors mention about it in materials and methods?
Line 219: Arabidopsis and Zea should be given with species name and should be italicized.
Line 219 and 220: Authors talk about 14 Moso bamboo CAMTA genes and 17 Moso bamboo CAMTA genes. However, they only found 11 CAMTA genes in Moso bamboo. Authors should rewrite these sentences for indicating CAMTA genes were paired with 14 and 17 times with Zea mays or Oryza sativa genome. Additionally, authors should calculate again Oryza sativa matches, are they 17 or 19?
Line 222, 223, 224: Arabidopsis and Zea should be given with species name and should be italicized.
Line 246: “PHCAMTA07/11 expression profile was” should be “PeCAMTA07 and PeCAMTA11 expression profiles were”.
Line 248: “PHCAMTA11 was more highly expressed in each tissue”. That is wrong. PeCAMTA11 was highly expressed on in Panicle tissue. Authors can say that PeCAMTA07 is highly expressed in each tissue.
Line 254: “the expression patterns” is a start of a new sentence and must start with a capital letter.
Line 258: Mao bamboo or moso bamboo? Additionally, authors left that sentence half, they should complete the sentence.
Line 261: “most of the genes” is a start of a new sentence and must start with a capital letter.
Line 264: “CAMTA” should be “CAMTAs”
Line 265: “PHCMATA” should be “PeCAMTAs”
Line 266: Expression profiles of CAMTA genes under abiotic stress conditions is the only experimental part of this study. Why do authors not give their cold and heat stress results and they only give drought stress results? Authors should give their all results here.

Line 415: Reference should be corrected.
Line 438: Reference should be corrected. Reference DOI is missing.
Line 532: Reference volume, issue and page number are missing.

Figure 3 and 4: Why Figure 3 contains 9 scaffolds in total, but Figure 4 contains 10 scaffolds in total? Why scaffold22 does not exist in Figure 3 if it carries a CAMTA gene?
Figure 5: “genens” should be “genes”

---

## Round 0.2 · Minor Revisions

Dear Authors

The manuscript needs a minor revision to be reconsidered for publication. The authors are invited to revise the paper considering all the suggestions made by the reviewer. Please note that requested changes are required for publication.
With Thanks

·

Basic reporting

1. In the rebuttal letter, authors have mentioned about the modifications according to the reviewers’ comments. I appreciate it but most of the lines mentioned in the rebuttal letter describe something else in the “revised version”.
Example: please check the rebuttal letter regarding the suggested sentences in the manuscript “c) The expression of Moso bamboo CAMTA was higher…………………………., d) For example, PHCAMTA07 was highly expressed……………………………….. and g) “The PheE2F/DP promoter in response to drought………………………….”.

2. Why different styles have been followed for different subheadings? Rectify it.
Some headings are mentioned below for your reference
“Expression of PeCAMTA Genes in Responses to Cold, Drought and Heat Treatments”
“The role of PeCAMTA genes in different tissues and organs”
“Evolutionary Characterization of the PeCAMTA gene family”
In some sentence, each words are capitalized, somewhere the first word is capitalized while somewhere only two words are capitalized. Make it uniform.
3. Figure 8 and 9 are almost providing similar information. Is there any extra information available in figure 9? Hence I suggest that figure 9 can be included in supplementary file.

Experimental design

3. Figure 8 and 9 are almost providing similar information. Is there any extra information available in figure 9? Hence I suggest that figure 9 can be included in supplementary file.

Validity of the findings

No comment

Additional comments

Authors can modify the subheadings and figure numbering (if figure 9 is converted into supplementary file).

---

## Round 0.3 · Minor Revisions

Dear Authors

The manuscript still needs editing.

1. The authors should be consistent in using genus/species names rather than common names throughout the manuscript. As it is now, there are some instances of genus/species (title), common name (abstract), sometimes genus, sometimes genus/species, and sometimes common name in the remainder of the manuscript. Always a space between the genus/species "P.edulis." After you cite the genus/species, no need to continue to spell out the genus (i.e., Zea mays).

2. Please note that you always need a space before open parentheses "(".

3. Be sure to supply relevant citations and/or software location (i.e., Plant Care)."

Best Reagrds

·

Basic reporting

Authors have made all the required changes. The manuscript looks fine now.

Experimental design

Thorough experiments were conducted in a rigorous manner.

Validity of the findings

No comments

Additional comments

The manuscript has been corrected and can be considered for publication.

---

## Round 0.4 · accepted · Accept

Dear Authors

I am pleased to inform you that after the last round of revision, the manuscript has been improved a lot, and it can be accepted for publication.
Congratulations on the acceptance of your manuscript, and thank you for your
interest in submitting your work to PeerJ